# Recycling of Gas Phase Residual Dichloromethane by Hydrodechlorination: Regeneration of Deactivated Pd/C Catalysts

**Sichen Liu, María Martin-Martinez**, **María Ariadna Álvarez-Montero**,
**Alejandra Arevalo-Bastante, Juan José Rodriguez and Luisa María Gómez-Sainero** *

Departamento de Ingeniería Química, Facultad de Ciencias, Universidad Autónoma de Madrid, Cantoblanco, 28049 Madrid, Spain

* Correspondence: luisa.gomez@uam.es; Tel.: +34-91-497-6939

**Abstract:** Dichloromethane (DCM) is an important pollutant with very harmful effects on human health and the environment. Catalytic hydrodechlorination (HDC) is an environmentally friendly technology for its removal from gas streams; it avoids the formation of hazardous pollutants like dioxins and phosgene (produced by other techniques), and the products obtained can be reused in other industries. When compared to other precious metals, Pd/C catalyst exhibited a better catalytic activity. However, the catalyst showed a significant deactivation during the reaction. In this study, the oxidation state and particle size of Pd was monitored with time on stream in order to elucidate the transformations that the catalyst undergoes during HDC. The deactivation can be ascribed to the formation of a new $PdC_x$ phase during the first hour of reaction. Carbon atoms incorporated to Pd lattice come from (chloro)-hydrocarbons adsorbed in the metallic species, whose transformation is promoted by the HCl originating in the reaction. Nevertheless, the catalyst was regenerated by air flow treatment at 250 °C, recovering the catalyst more than 80% of initial DCM conversion.

**Keywords:** palladium; activated carbon; hydrodechlorination; deactivation; regeneration; palladium carbide

---

## 1. Introduction

Dichloromethane (DCM) has been widely used in the chemical industry, mainly as an organic solvent. However, due to its low boiling temperature (40 °C), DCM, as a volatile liquid compound, belongs to a group of organochlorinated pollutants which contributes to the destruction of the ozonosphere, the photochemical pollution and the greenhouse effect. On the other hand, organochlorinated compounds are very toxic and carcinogenic [1–5]. Therefore, finding effective technologies to remove chloromethanes from residual streams is necessary. The catalytic hydrodechlorination (HDC) is one of the most promising technologies to treat the residual chloromethanes due to its moderate operating conditions, using atmospheric pressure and relative low temperature. Moreover, the reaction products (non-chlorinated hydrocarbons), are much less hazardous than both the main reactives and the products that would be obtained by other techniques, e.g., production of $NO_x$ or dioxins by oxidative treatments of chlorinated compounds [6] and can be recycled for chemical or energetic purposes. Hence, the HDC technology presents more economic and environmental advantages than other technologies [7–13].

In recent years, the HDC of different chloromethanes have been investigated using heterogeneous catalysts based on different metals and supports [7,14–19]. Four precious metals, platinum, palladium, ruthenium and rhodium, have been more frequently employed as active phases in the literature.

Among them, palladium has been often considered the most efficient one due to its high catalytic activity [16,18]. In addition, it usually results in a higher selectivity to C2–C3 hydrocarbons (more valuable products than methane), when comparing to other metals like platinum [7,14,17]. The main reactions involved are depicted in Scheme 1. Gómez-Sainero et al. [17] reported, from a theoretical and experimental study, that, comparing to the other three mentioned precious metals, the palladium supported on activated carbon (Pd/C) used in the HDC of chloroform (TCM) and DCM is more suitable to produce C2 products, which are valuable hydrocarbons in various chemical industries such as the plastics, pharmaceutical and fine chemical sectors.

$$\text{I: } 2CH_2Cl_2 + 4H_2 \leftrightarrow 2CH_4 + 4HCl$$

$$\text{II: } 2CH_2Cl_2 + 3H_2 \leftrightarrow C_2H_6 + 4HCl$$

$$\text{III: } 3CH_2Cl_2 + 4H_2 \leftrightarrow C_3H_8 + 6HCl$$

**Scheme 1.** Hydrodechlorination reactions of dichloromethane to methane, ethane and propane.

However, Pd/C has demonstrated a significant deactivation during the HDC of chlorinated compounds in liquid and gas phase. Simagina et al. [20] observed that Pd/C deactivated in the liquid phase HDC of chlorobenzene (CB) due to the HCl produced, which destroyed small Pd particles leading to a loss of activity. The formation of HCl was also claimed by Concibido et al. [21] as the main reason of Pd/C deactivation during the HDC of tetrachloroethylene (TTCE). Moon et al. [22] and Zheng et al. [23] observed sintering of Pd particles in a Pd/C catalyst, leading to catalyst deactivation in the HDC of chloropentafluoroethane ($CF_3CF_2Cl$). Zheng et al. [23] also used Pd/C in the HDC of dichlorodifluoromethane ($CF_2Cl_2$). In this case, the deactivation causes were attributed to the carbonaceous deposits on the catalyst and the formation of palladium carbide ($PdC_x$) which was also observed in other studies of the HDC reaction [24–26]. In a previous study [14], the catalytic activity of Pd/C and Pt/C in the HDC of DCM and TCM was compared. Despite showing better initial catalytic activity than Pt/C, Pd/C catalyst showed a significant deactivation, especially in the HDC of DCM, which was mainly ascribed to the poisoning of the active centers by the reactant, leading to the formation of $PdC_x$ phase by the dissociative adsorption of DCM. Furthermore, sintering of Pd particles, associative adsorption of reactants and reaction products and formation of carbonaceous deposits on the Pd phase were also found to strongly contribute to the deactivation of Pd/C.

The regeneration of deactivated catalysts used in the HDC of chlorinated compounds has been investigated in various studies [27–30]; however, to our knowledge, results on regeneration of Pd/C catalysts for the HDC of chloromethanes are not available in the literature. The methods of regeneration could be classified into in situ gas flow regeneration or regeneration by washing. The in-situ regeneration treatments seem to be more applicable due to their relatively easy manipulation. $H_2$, air, and even inert gases like argon are mainly employed to regenerate deactivated catalysts in HDC reaction [27–30]. The regeneration effects of each gas mainly depend on the different deactivation causes and the different metals and supports used in the HDC catalysts. González et al. [29] successfully removed carbonaceous deposits on the Pd phase of $Pd/TiO_2$ used in the HDC of DCM, TTCE and TCM by air flow, heating at temperatures lower than 400 °C. Moreover, an argon-flow could also remove the carbonaceous deposits generated on the surface of $Pd/TiO_2$ during the HDC of carbon tetrachloride ($CCl_4$) [30]. Legawiec-Jarzma et al. [27] employed the same support, $Al_2O_3$, combined with Pd-Pt, in the HDC of $CCl_4$, finding that an $H_2$ flow could not be used to remove the carbonaceous deposits accumulated on the catalyst and regenerate it. Ordoñez et al. [28] tested air and $H_2$ to regenerate the $Pd/Al_2O_3$ used in the HDC of TTCE, whose deactivation was ascribed to the poisoning by hydrogen chloride (HCl) and the formation of carbonaceous deposits on the catalyst. Both gas flows (air and $H_2$) resulted as not strong enough to completely recover the catalytic activity of $Pd/Al_2O_3$. Besides, the catalyst deactivation was faster after the regeneration. He also tried different washing methods, such as washing the deactivated catalyst using organic solvents (toluene, tetrahydrofurane and DCM) or with an ammonia solution. However, neither of these treatments were effective. On the

other hand, Concibido et al. [21] used MeOH, water and a mixture of MeOH and water to wash the deactivated Pd/C catalyst used in the HDC of TTCE, finding that the washed catalyst recovered its initial catalytic activity.

The aim of this work was firstly to conduct a rigorous study analyzing the evolution of the physico-chemical properties of the catalyst during the gas phase HDC of DCM, in order to investigate the causes of Pd/C deactivation. Hereafter, the objective was to find a feasible method to regenerate the used catalyst.

## 2. Results and Discussion

### 2.1. Evolution of Catalytic Activity

Four DCM HDC experiments of different reaction duration (5 h, 40 h, 80 h and 120 h) were performed with Pd/C, using a space-time ($\tau$) of 1.73 kg h mol$^{-1}$ and a reaction temperature of 250 °C, in order to determine the modifications suffered by the catalyst along the reaction. The characterization results of these samples are analyzed in Section 2.2. The experimental conditions used in these HDC experiments are described in detail in Section 3.3. Results of conversion and selectivities to the different reaction products obtained at the end of each experiment are summarized in Table 1. In addition, Figure 1 shows the evolution of the conversion and selectivities obtained in the experiment performed at the longest reaction period of 120 h. As can be seen in Table 1, DCM conversion began to decrease after the first 5 h of reaction, revealing the existence of some extent of catalyst deactivation. After 120 h of reaction (Figure 1), DCM still maintained a relatively high conversion (81.7%), suggesting that, at the operating conditions used, most of the catalyst might not be deactivated yet. On the other hand, the selectivity to the main products did not significantly change during the reaction. This behavior was also observed in another study of HDC of DCM with a similar distribution of reaction products [17].

**Table 1.** Catalytic activity at the end of the experiments performed at different times on stream.

| Duration (h) | Conversion (%) | Selectivity (%) | | | |
| --- | --- | --- | --- | --- | --- |
| | DCM | CH$_4$ | C$_2$H$_6$ | C$_3$H$_8$ | MCM |
| 0 | 100.0 | 77.8 | 16.5 | 1.1 | 4.6 |
| 5 | 98.0 | 78.4 | 13.4 | 1.0 | 7.1 |
| 40 | 96.8 | 78.3 | 14.0 | 1.0 | 6.7 |
| 80 | 90.5 | 78.5 | 13.4 | 1.0 | 7.1 |
| 120 | 81.7 | 78.5 | 13.1 | 1.1 | 7.3 |

With the aim of evaluating the efficiency of thermal regeneration treatments applied to Pd/C catalysts deactivated with chloromethanes, we performed a DCM HDC experiment using a lower space-time (0.40 kg h mol$^{-1}$) to obtain a stronger deactivation of the catalyst, and maintained it during 200 h on stream. After this period, Pd/C lost more than 50% of the initial catalytic activity and was regenerated by treatment with a 50 mL min$^{-1}$ air flow for 12 h at 250 °C. The experimental conditions are presented in detail in Section 3.3.

The evolution of catalytic activity at a space-time of 0.40 kg h mol$^{-1}$ is shown in Figure 2. As expected, comparing to the previous essay, the deactivation of the catalyst was more pronounced due to the lower number of active sites, and the catalysts deactivated quickly in the first hour (Figure 2). After the regeneration treatment, 80% of the initial conversion was recovered. However, the activity decay of regenerated catalyst was lower, and after 70 h of reaction, regenerated and the fresh catalyst exhibited a similar conversion around 40%. On the other hand, the selectivity to the main reaction products remained similar to the distribution shown by the fresh catalyst. Therefore, regeneration by air flow seems to be an efficient method for recovering the catalytic activity of used Pd/C. To understand the catalyst deactivation causes and the effect of air treatment in the regeneration of the catalyst,

the physico-chemical properties of the catalyst will be studied by means of several characterization techniques in Section 2.2 and their experimental conditions will be illustrated in Section 3.2.

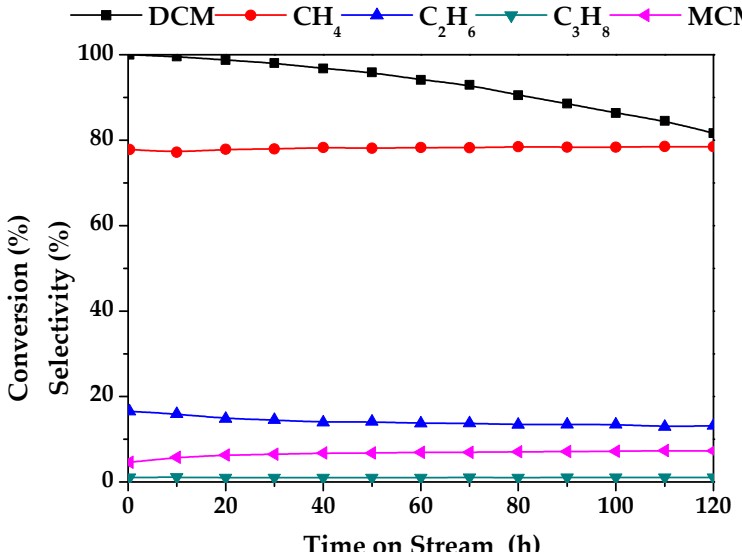

**Figure 1.** Evolution of dichloromethane (DCM) conversion and selectivity to reaction products upon time on stream in the hydrodechlorination (HDC) of DCM during 120 h ($\tau$ = 1.73 kg h mol$^{-1}$, $T_{reaction}$ = 250 °C).

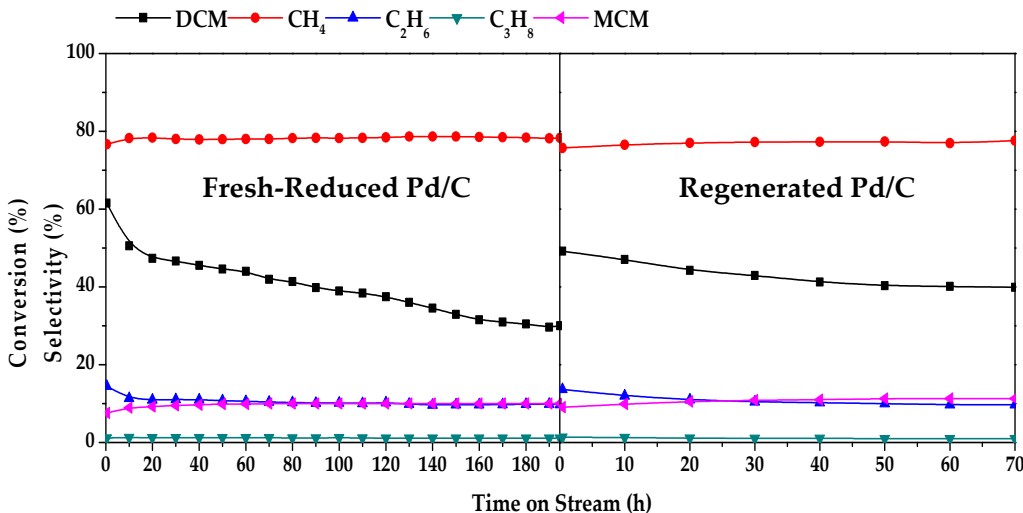

**Figure 2.** Evolution of DCM conversion and selectivity to reaction products upon time on stream in the HDC of DCM with the fresh-reduced and regenerated Pd/C (the catalyst regeneration in air flow 50 mL min$^{-1}$, 250 °C for 12 h, $\tau$ = 0.4 kg h mol$^{-1}$).

## 2.2. Characterization Results

The surface areas and micropore volumes of the fresh and used Pd/C catalysts at different reaction times were analyzed by $N_2$ adsorption–desorption at −196 °C. The surface areas were calculated by Brunauer-Emmett-Teller (BET) equation. The results are summarized in Table 2.

No significant changes on catalyst surface area during the reaction were observed, as can be seen in Table 2. The difference between the fresh-reduced catalyst (1161 m$^2$ g$^{-1}$) and the used catalyst after the final reaction (1114 m$^2$ g$^{-1}$) was only 4%. Similar results were observed in the micropore volumes. Blockage of the porous structure did not seem to occur during reaction.

**Table 2.** BET Surface area and micropore volume of the catalysts.

| Catalyst | BET Surface Area ($m^2\ g^{-1}$) | Micropore Volume ($cm^3\ g^{-1}$) |
| --- | --- | --- |
| Fresh-reduced Pd/C | 1161 | 0.53 |
| Pd/C 5 h | 1155 | 0.53 |
| Pd/C 40 h | 1129 | 0.52 |
| Pd/C 80 h | 1132 | 0.52 |
| Pd/C 120 h | 1114 | 0.51 |

Figure 3 shows some Transmission Electron Microscopy (TEM) images and the distributions of the palladium particles sizes in the samples obtained for the four experiments at different reaction times. The distributions of Pd particles size were obtained by counting more than 400 particles on each sample. TEM images show that Pd particles were well and similarly dispersed in all the samples, with small particles sizes in all cases (between 1.0 nm and 2.5 nm). No significant rise of palladium size was observed after the HDC reaction, which indicates that the deactivation of the catalyst in the HDC of DCM cannot be attributed to sintering of metallic particles.

Figure 4 shows the X-ray diffraction (XRD) patterns of the Pd/C catalysts. No characteristic peaks of metallic palladium were observed, which can be attributed to the very small particle size of palladium supported on the activated carbon, as it was seen by TEM (Figure 3). From XRD results, we could confirm that Pd particles did not sinter during the HDC reaction, according to the results obtained by TEM (Figure 3).

Table 3 summarizes the catalyst surface composition (Pd and Cl), as determined by X-ray photo-electron spectroscopy (XPS) and the relative proportions of the different species of these two elements in the fresh-reduced, used at different reaction times, and regenerated catalysts. Figures 5 and 6 show the XPS profiles of the catalysts in the Pd 3d region. The Pd 3d region was deconvoluted using an area ratio of 0.66, corresponding at d orbital, and a doublet separation of ~5.26 eV. In all cases, the Pd 3d region spectra presented a doublet corresponding to Pd $3d_{5/2}$ and Pd $3d_{3/2}$. For the fresh catalyst, a peak was located at a binding energy value around 334.9 eV, corresponding to metallic palladium ($Pd^0$), and another peak was centered at around 336.8 eV, attributed to electro-deficient palladium ($Pd^{n+}$) [31,32]. The formation of the $Pd^{n+}$ in reduced Pd catalysts was also observed in other studies [33,34]. When the XPS profiles of the used catalysts at different times on stream were deconvoluted, a new phase appeared around 336.1–335.5 eV (Figures 5 and 6), which can be attributed to $PdC_x$, according to the NIST database and Setiawan et al. [35] who identified the formation of this specie at a binding energy of the $Pd3d_{5/2}$ peak at around 335.9 eV. As can be seen in Table 3, the proportion of $PdC_x$ phase mainly rose during the first 5 h of reaction, though a slight increase is observed up to 40 h. Beyond this time the amount remains constant. A great reduction of the relative atomic proportion of the $Pd^0$ was noticed, which decreased from 52.5% to 11.5%. Meanwhile, the $Pd^{n+}$ relative atomic proportion rose slightly from 47.5% to 50.7%. The proportion of $PdC_x$ mainly increases to the detriment of the $Pd^0$, which suggests that $Pd^0$ is the main specie involved in the formation of Pd carbide during the HDC reaction. It seems that during the first 5 h, significant changes occurred in the catalyst surface, which might be attributed to a modification of the nature of the catalyst active centers. On the other hand, as can be seen in Table 3, the chlorine proportion on the surface increased from 0.07% to 0.33% during the first 5 h of reaction. In particular, the inorganic chlorine proportion also increased in the first 5 h. Because of the formation of $PdC_x$, the $Pd^{n+}$ deconvoluted peaks displaced to around 338 eV. According to the literature [34,36], $Pd^{n+}$ come from the interaction of metallic Pd with chlorides. The electron-donor Pd atoms interact with the neighboring electrophilic $H^+$, which lead to the formation of electro-deficient palladium ($Pd^{n+}$). The interaction is stabilized by the $Cl^-$ present on the catalyst, according to the following mechanism:

$$Pd^0 + H^+ + Cl^- \rightarrow Pd^{n+} \cdots H^{(1-n)+} \cdots Cl^- \tag{1}$$

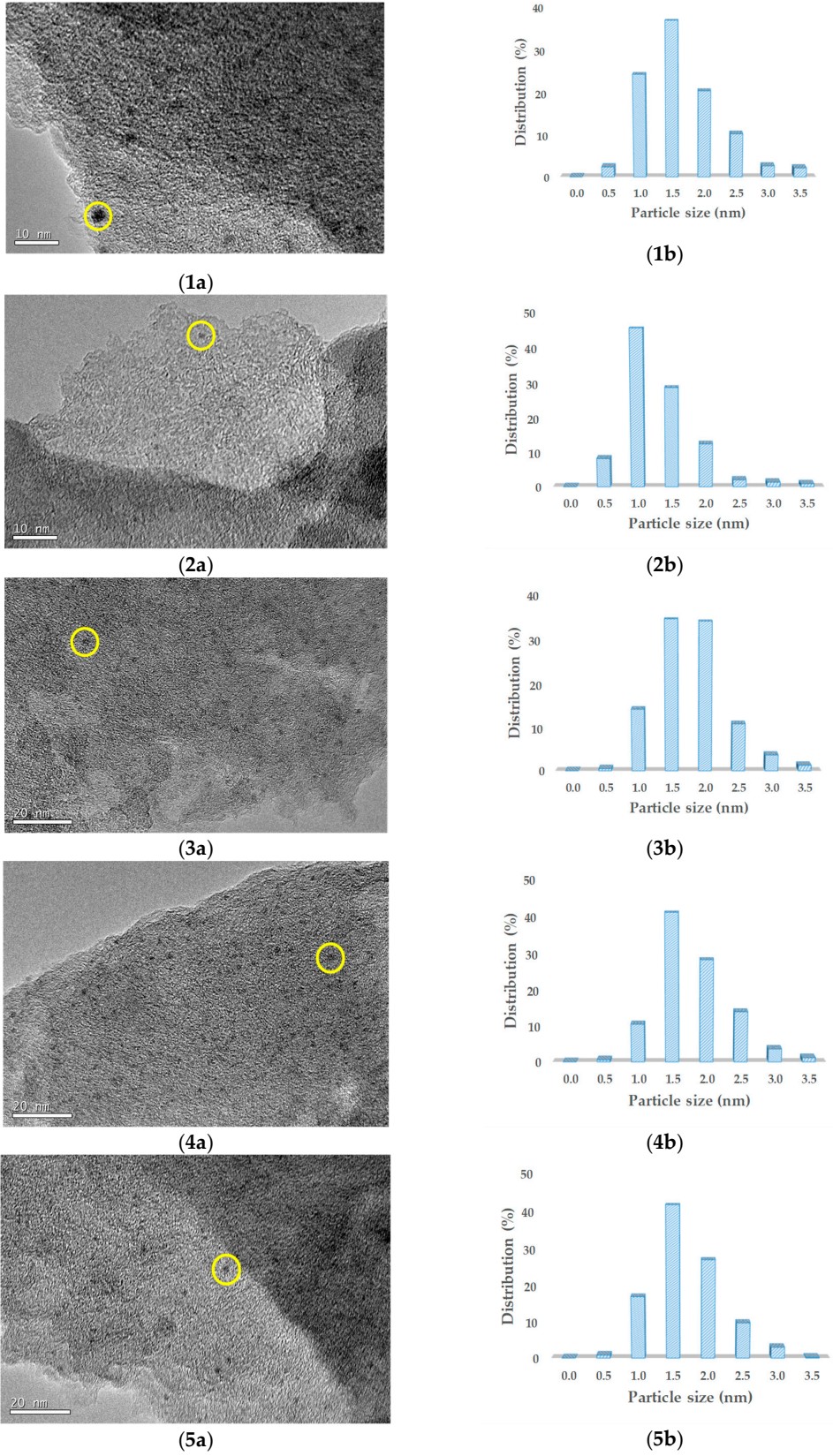

**Figure 3.** TEM images (**a**) and palladium particle size distribution (**b**) of the fresh-reduced (**1**), 5 h used (**2**), 40 h used (**3**), 80 h used (**4**), and 120 h used (**5**) Pd/C catalysts.

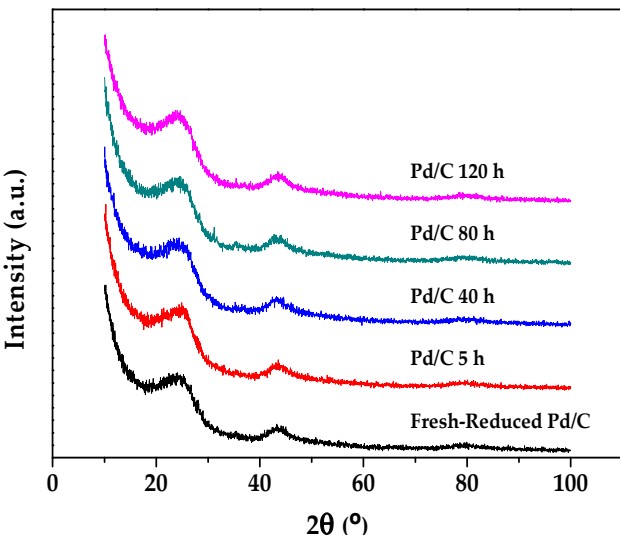

**Figure 4.** X-ray diffraction (XRD) patterns of Pd/C used in the HDC of DCM for 5 h, 40 h, 80 h and 120 h.

**Table 3.** XPS results in relative atomic composition in the two studies (experiments at different time on stream ($\tau$ = 1.73 kg h mol$^{-1}$); deactivation–regeneration study ($\tau$ = 0.40 kg h mol$^{-1}$)).

| | Catalyst | Surface Composition | | Surface Species | | | | |
|---|---|---|---|---|---|---|---|---|
| | | $Pd_{XPS}$ (%) | $Cl_{XPS}$ (%) | $Pd^0$ (%) | $Pd^{n+}$ (%) | $PdC_x$ (%) | $Cl_{inorg}$ (%) | $Cl_{org}$ [1] (%) |
| Experiments at different time on stream | Fresh-Reduced Pd/C | 0.10 | 0.07 | 52.5 | 47.5 | 0.0 | 63.2 | 36.8 |
| | Used Pd/C 5 h | 0.08 | 0.33 | 11.5 | 50.7 | 37.8 | 69.4 | 30.6 |
| | Used Pd/C 40 h | 0.10 | 0.10 | 11.9 | 44.1 | 44.0 | 57.9 | 42.1 |
| | Used Pd/C 80 h | 0.08 | 0.09 | 11.0 | 43.3 | 45.7 | 49.1 | 50.9 |
| | Used Pd/C 120 h | 0.09 | 0.13 | 11.1 | 46.1 | 42.8 | 51.2 | 48.8 |
| Deactivation–regeneration study | Deactivated Pd/C 200 h | 0.11 | 0.12 | 10.0 | 14.1 | 75.9 | 51.5 | 48.5 |
| | Regenerated Pd/C by Air | 0.37 | 0.20 | 31.8 | 65.7 | 2.5 | 46.7 | 53.3 |

[1] Referred to chlorine from organic species.

The significant increase of chlorine concentration in the catalyst surface exposed above suggests that because of the high proportion of HCl produced as a consequence of the HDC reaction (Scheme 1), a high proportion of HCl was adsorbed on the activated carbon surface, in the form of Cl$^-$ and H$^+$ by which the Pd$^0$ is converted to Pd$^{n+}$ (Equation (1)). Therefore, the surface Pd$^0$ proportion decreased (Table 3). However, only a small increment of the relative atomic proportion of Pd$^{n+}$ was observed (Table 3). As can be seen in previous works [14,37], Pd$^{n+}$ promotes the irreversible chemisorption of DCM and reaction intermediates. This favors the incorporation of carbon atoms coming from these species into the Pd lattice, leading to the formation of the new PdC$_x$ phase. The process is enhanced by the high capability of Pd$^{n+}$ to dissociate H$_2$, needed to stabilize the chlorides extracted from the chloromethanes.

After those first 5 h, the chlorine atomic proportion on the activated carbon surface decreased again (Table 3, experiments at different time on stream), as does the inorganic chlorine proportion. This could be interpreted by the irreversible formation of PdC$_x$, which might weaken the interaction between the Pd$^{n+}$ with H$^+$ and Cl$^-$, because part of Pd$^{n+}$ has already been converted to PdC$_x$, which should be more stable. Thus, with a weak interaction with palladium atoms, the HCl might desorb more easily from the activated carbon surface. These results suggest that the inorganic chlorine (HCl) played an important role in the generation of Pd$^{n+}$, which might chemisorb DCM and products, and finally be stabilized by the formation of the PdC$_x$.

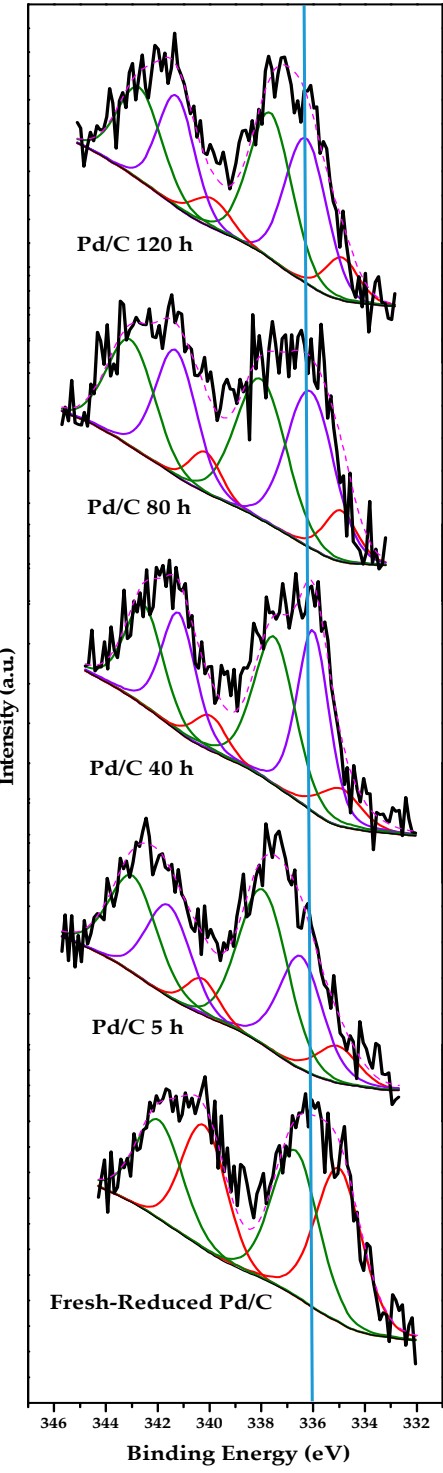

**Figure 5.** XPS patterns in the Pd 3d region of the Pd/C catalyst after the experiments performed at different times on stream ($\tau$ = 1.73 kg h mol$^{-1}$) (the overall Pd 3d spectra (black); Pd$^0$ spectra (red); Pd$^{n+}$ spectra (green); PdC$_x$ spectra (violet); overall fitting line spectra (dash); the blue vertical line indicates the position of the peak of PdC$_x$ spectra).

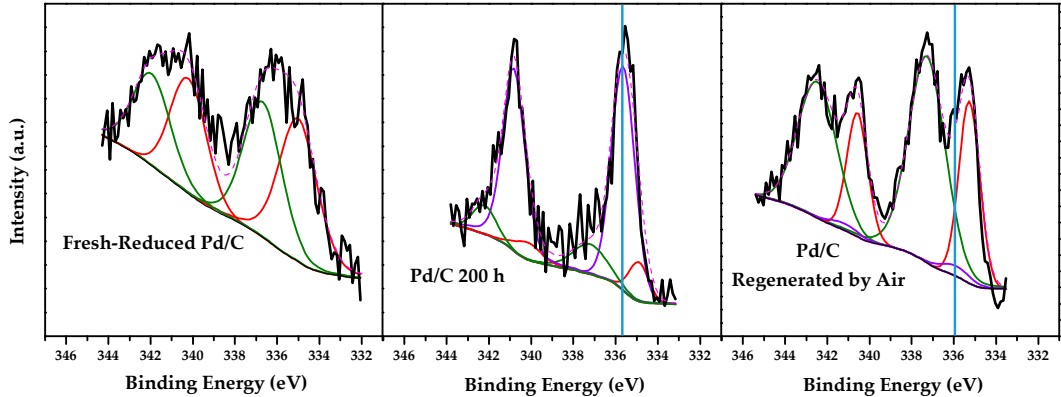

**Figure 6.** XPS patterns in the Pd 3d region of the Pd/C catalysts from the deactivation–regeneration study ($\tau = 0.40$ kg h mol$^{-1}$) (the overall Pd 3d spectra (black); Pd$^0$ spectra (red); Pd$^{n+}$ spectra (green); PdC$_x$ spectra (violet); overall fitting line spectra (dash); the blue vertical line indicates the position of the peak of PdC$_x$ spectra).

The results obtained for the catalyst after the strong deactivation suffered by performing the experiment at a lower space-time are in agreement with those explained above. After 200 h of operation at a space-time of 0.40 kg h mol$^{-1}$, 75.9% of Pd appears as PdC$_x$. After the regeneration treatment by air, the proportion of PdC$_x$ dropped to a very low value, 2.5%, resulting in the recovery of catalytic activity (Figure 2). This supports that formation of PdC$_x$ is the main cause of catalyst deactivation and that treatment by air flow could successfully remove carbonaceous material from Pd.

However, the regenerated catalyst shows a greater amount of Pd in the outer surface of the catalysts and a higher ratio of oxidized Pd$^{n+}$ (65.7%), which could explain the slight differences in the activity when compared to the fresh catalyst. A little increase of the overall chlorine was observed in the regenerated catalyst (Table 3), which can be related to the higher amount of Pd$^{n+}$.

With the aim of confirming the formation of PdC$_x$ phase by XRD characterization, we prepared a Pd/C catalyst with the same composition and Pd precursor but with bigger Pd particles, which was tested in the HDC reaction and regenerated at the same operating conditions as those presented in Figure 2. As can be seen in Figure S1, after deactivation, the XRD pattern of the deactivated catalyst shows clear peaks corresponding to PdC$_x$ phase, which disappear after regeneration, recovering the diffraction peaks of metallic Pd.

## 3. Materials and Methods

### 3.1. Catalysts Preparation

Incipient wetness impregnation method was used for preparing the Pd/C catalyst with a nominal load of Pd of 1 wt%. A commercial activated carbon (Erkimia S.A., Barcelona, Spain) was employed as catalytic support (external diameter of the carbon particle between 0.25 and 0.50 mm). The exact volume of the aqueous solution of PdCl$_2$ (supplied by Sigma-Aldrich, Madrid, Spain) used for the impregnation was prepared according to the pore volume of the activated carbon. After drying up to 100 °C, the catalyst was reduced in situ at 250 °C using a continuous 50 mL min$^{-1}$ H$_2$ flow (minimum purity: 99.999%, supplied by Praxair, Madrid, Spain) for 2 h.

### 3.2. Catalyst Characterization

N$_2$ adsorption–desorption at −196 °C analyses (Micromeritics Tristar II 3020) were performed to characterize the porous structure of the catalysts. The samples were previously cleaned by an outgassing treatment at 150 °C for 12 h, with a 10$^{-3}$ Torr residual pressure, using a Micromeritics VacPrep 061. The BET equation was used to calculate the surface areas and the t-method was applied to obtain the micropore volumes.

TEM images were obtained using a JEOL JEM-2100F microscope operating at 200 kV with a 0.19 nm point resolution. It employs a high-angle annular dark field (HAADF) detector, a 2 k × 2 k ULTRASCAN 1000 CCD camera, and an Oxford Instruments INCA Energy TEM 250 for chemical analysis by Energy Dispersive X-ray Spectroscopy (XEDS). The samples were previously dispersed in ethanol and dropped onto holey carbon-coated Cu grids.

XRD was applied to analyze the crystalline structure of the catalysts, employing a X'Pert PRO Panalytical Diffractometer. CuKα monochromatic radiation (α = 0.15406 nm) was used to scan the powdered sample using a Ge Mono filter. The scanning range was 10–100° (2θ) and the scan step size was 0.0334. The resulting signals were collected every 200 seconds with an X'Celerator RTMS detector.

XPS was used to characterize the surface composition of the catalysts, using a Physical Electronics 5700 C Multitechnique System applying MgKα radiation ($h_m$ = 1253.6 eV). The elements present were determined by general XPS spectra scanning up to 1200 eV binding energy (BE). C1s peak (284.6 eV) was used to correct the BE changes by sample charging. The BE of the Cl 2p and Pd 3d core levels, and the full width at half maximum (FWHM) values were used to estimate the chemical state of Cl and Pd on the catalyst surface. The produced spectra were deconvoluted using mixed Gaussian–Lorentzian functions by a least-squares method using Multipak v8.2 software (ULVAC-PHI, Inc., Chigasaki, Kanagawa, Japan)—a specific program to handle XPS data. The elements' atomic compositions were obtained from the relative summit areas of the core level curves using Wagner sensitivity factors [38].

### 3.3. Catalytic Activity Experiments

The HDC experiments were performed in a continuous flow reaction system described before [31], consisting in a quartz fixed bed micro-reactor of a 9.5 mm i.d., connected to a gas-chromatograph (column Plot Fused Silica 60 m × 0.53 mm ID, Bruker, Madrid, Spain) using an FID detector to analyze the evolution of main reagents and reaction products.

For the two different studies, the common experiments' conditions are: operation at atmospheric pressure; employing a total flow rate of 100 N $cm^3min^{-1}$; a reagents molar ratio of 100 ($H_2$/DCM); a reaction temperature of 250 °C. The inlet DCM concentration for both studies was 1000 ppmv, which was prepared by mixing a DCM/$N_2$ commercial mixture and $N_2$ in the appropriate ratios (both gases supplied by Praxair, Madrid, Spain). According to a previous study, under these conditions, transfer limitations might be discarded [39]. The catalysts were reduced in situ under continuous $H_2$ flow at 300 °C for 90 minutes prior to the HDC experiments. Their activity was evaluated in terms of DCM conversion and selectivity to different reaction products. In the first study (experiments performed at different times on stream: 5 h, 40 h, 80 h, 120 h), the space-time of the catalysts used in each of the 4 experiments was 1.73 kg h $mol^{-1}$. In the second study (deactivation–regeneration study), the space-time of the catalyst was 0.40 kg h $mol^{-1}$, with a relatively long time on stream: 200 h. After the first 200 h, the used catalyst was regenerated by an in-situ air flow of 50 mL $min^{-1}$ at 250 °C during 12 h. Then, the regenerated efficiency was evaluated by another 24 h HDC reaction of DCM using the regenerated catalyst.

The experimental results were reproducible with less than 5% error.

## 4. Conclusions

The deactivation of the Pd/C catalyst in the catalytic hydrodechlorination of dichloromethane was mainly ascribed to the irreversible chemisorption of reactants and reaction products in the active centers, which resulted in the formation of a new palladium carbide phase. The HCl produced in the reaction is reversibly adsorbed on the activated carbon surface in the first 5 h of HDC. This might be the key of the generation of new electro-deficient palladium, which would be subsequently stabilized by the formation of $PdC_x$. The air flow regeneration treatment successfully recovers more than 80% of the initial DCM conversion. Therefore, it appears as a suitable method to recover the catalytic activity of the deactivated Pd/C catalyst used in the HDC of DCM.

**Supplementary Materials:** The following are available online at http://www.mdpi.com/2073-4344/9/9/733/s1, Figure S1: (**a**) XRD patterns of Pd/C (containing bigger Pd particles) used in the deactivation-regeneration study, (**b**) partial detail (diffraction peaks of metallic Pd (black and blue); diffraction peaks of PdC$_x$ (red)).

**Author Contributions:** Conceptualization, J.J.R. and L.M.G.-S.; investigation, S.L., M.M.-M. and A.A.-B.; methodology, S.L., M.M.-M., A.A.-B. and L.M.G.-S.; data analysis, S.L., M.M.-M., M.A.A.-M., J.J.R. and L.M.G.-S.; supervision, M.M.-M., M.A.A.-M., J.J.R. and L.M.G.-S.; writing—original draft preparation, S.L.; writing—review and editing, S.L., M.M.-M., M.A.A.-M., J.J.R. and L.M.G.

**Funding:** The authors gratefully acknowledge financial support from FEDER/Ministerio de Ciencia, Innovación y Universidades—Agencia Estatal de Investigación/CTM2017-85498-R. S. Liu acknowledges MINECO for his research grant/PRE2018-084424. M. Martin-Martinez acknowledges the postdoc grant 2017-T2/AMB-5668 from Comunidad de Madrid, Programme "Atracción de talent investigador".

**Conflicts of Interest:** The authors declare no conflict of interest.

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
