# Peer review of "Recycling of Gas Phase Residual Dichloromethane by Hydrodechlorination: Regeneration of Deactivated Pd/C Catalysts"

_catalysts, doi:10.3390/catal9090733_

Round 1
Reviewer 1 Report
The authors report the catalytic behavior and characterize Pd nanoparticles with various HDC reaction times. Regeneration of deactivated Pd/C catalysts is also studied. The Pd nanoparticles are examined by TEM, XRD, and XPS. The authors show that the PdCx phase appears after the HDC reaction. The catalytic behavior is well explained by the deconvolution results of XPS analysis. The air flow regeneration is effective to recover the activity of deactivated Pd/C catalysts from DCM conversion. The results is interesting and significant. However, some points need to be clarified before the manuscript could be considered for publication.
1. Page 7 Figure 4. I think Figure 4(a) and 4(b) are very similar. I suggest the authors to remove one of them. Besides, the legends in Figure 4(b) are all the same for the top four lines. It shall be corrected.
2. Page 7 Table 3. I have difficulty in reading this table because the composition and its deconvolution percentage are mixed together. The authors shall find a way to make the table more clearly. Besides, the sum of each deconvolution percentage is not exactly 100%. The authors shall also correct it.
3. Page 7 Table 3. What is organic chlorine? The authors did not mention it in the text.
4. Page 9 Figure 5 and 6. The caption for line in deconvoluted Pdn+ spectra is not black. It is green. The fitted dashed line shall be added to the caption.
5. Page 9 Figure 5 and 6. The way how the XPS spectra is fitted shall be addressed more clearly. The area ratio used for fitting the spectrum with the Pd 3d5/2 and 3d3/2 peaks shall be added in the text.
6. In this study, the formation of PdCx phase is observed by XPS fitting. Is it possible to justify it by selective area diffraction from TEM?
Reviewer 2 Report
Manuscript ID: catalysts-579277
Gomez-Sainero et al. reported on the deactivation system of the Pd nanoparticles in the catalytic hydrodechlorination (HDC) reaction of dichloromethane (DCM). They found the formation of a palladium carbide (PdCx) species during the first hours of reaction. The inorganic HCl played an important role for generation of Pdn+, which promotes the chemisorption of DCM. The Pd catalysts during the reaction at the different times were characterized by BET, TEM, and XRD measurements, indicating no significant change of the catalysts. The XPS analysis of the Pd catalysts exhibited the formation of the PdCx and the regeneration by inexpensive air flow. This method is suitable for regeneration of the deactivated Pd catalysts in the HDC of DCM. This paper is recommended to be published in Catalysts, but a few minor revisions seem to be considered by addressing following comments.
Although the authors prepared the fresh-reduced Pd catalysts using H2 flow, the regeneration of the catalysts was used by air flow. The XPS peaks of fresh-reduced Pd and regenerated Pd catalysts show similar peaks including metallic Pd0 and electro-deficient Pdn+ spiecies. If the authors use H2 flow for the regeneration of the Pd catalyst, is it possible to improve a higher percentage than the previous conversion of 80%? The authors described a couple of TEM images at different reaction times, as shown in Figure 3. After air flow regeneration, they should confirm the sizes of the Pd nanopaticles again using TEM measurements. One possibility is to observe more dispersed partciles than the fresh-reduced palladium particles.
